# A multicolor and ratiometric fluorescent sensing platform for metal ions based on arene–metal-ion contact

Anna Kanegae[1,5], Yusuke Takata[1,5], Ippei Takashima[1,4], Shohei Uchinomiya[1], Ryosuke Kawagoe[1], Kazuteru Usui [1], Akira Yamashita[1], Jirarut Wongkongkatep[2], Manabu Sugimoto[3] & Akio Ojida [1✉]

Despite continuous and active development of fluorescent metal-ion probes, their molecular design for ratiometric detection is restricted by the limited choice of available sensing mechanisms. Here we present a multicolor and ratiometric fluorescent sensing platform for metal ions based on the interaction between the metal ion and the aromatic ring of a fluorophore (arene–metal-ion, AM, coordination). Our molecular design provided the probes possessing a 1,9-bis(2′-pyridyl)-2,5,8-triazanonane as a flexible metal ion binding unit attached to a tricyclic fluorophore. This architecture allows to sense various metal ions, such as Zn(II), Cu(II), Cd(II), Ag(I), and Hg(II) with emission red-shifts. We showed that this probe design is applicable to a series of tricyclic fluorophores, which allow ratiometric detection of the metal ions from the blue to the near-infrared wavelengths. X-ray crystallography and theoretical calculations indicate that the coordinated metal ion has van der Waals contact with the fluorophore, perturbing the dye's electronic structure and ring conformation to induce the emission red-shift. A set of the probes was useful for the differential sensing of eight metal ions in a one-pot single titration via principal component analysis. We also demonstrate that a xanthene fluorophore is applicable to the ratiometric imaging of metal ions under live-cell conditions.

[1] Graduate School of Pharmaceutical Sciences, Kyushu University, Fukuoka, Japan. [2] Department of Biotechnology, Faculty of Science, Mahidol University, Bangkok, Thailand. [3] Department of Science and Technology, Graduate School of Kumamoto University, Kumamoto, Japan. [4] Present address: Laboratory of Bioorganic & Natural Products Chemistry, Kobe Pharmaceutical University, Kobe, Japan. [5] These authors contributed equally: Anna Kanegae, Yusuke Takata. ✉email: ojida@phar.kyushu-u.ac.jp

Fluorescent molecular probes are essential research tools in many fields as they can provide sensitive and selective detection of chemical species in real time. Among their sensing targets, metal ions have attracted special interest because of their significant biological effects. For harmful, non-essential heavy metal ions (mercury, cadmium, lead, etc.), fluorescent probes serve as tools to understand their biological activities in animals and plants[1], which occur as a result of unwanted uptake from the environment. Indeed, the detection of these harmful metal ions in organisms has been the subject of biological research for the last 30 years, and extensive studies are still being done to elucidate the various biological mechanisms associated with their toxic effects[2,3]. Meanwhile, monitoring toxic metal ion contamination in the environment is also an important task for preserving global health[4]. For biologically relevant metal ions (calcium, zinc, copper, iron, etc.), detection of their localization and stimuli-dependent concentration changes in live-cell and tissue contexts facilitates understanding of their physiological and pathological roles[5–7].

Fluorescent probes for metal ions are typically designed by joining a coordinating ligand unit to a luminescent sensing unit[8]. The design of ligands with high selectivity and tuned binding for a target metal ion has been well-developed as a result of the knowledge accumulated by years of coordination chemistry and supramolecular chemistry research. Meanwhile, exploitation of new mechanisms for metal ion sensing, which transduce the metal ion binding event into a fluorescence signal change, is still a challenging task[9,10]. The conventional sensing mechanism used for sensing metal ions is PET (photoinduced electron transfer)[11]. AIE (aggregation-induced emission)[12] and chemical reaction-based mechanisms (i.e., activity-based sensing)[9,10,13] have also been exploited for metal ion sensing in recent years. In addition to these off–on-type fluorescence sensing mechanisms, ICT (intramolecular charge transfer)[14] and FRET (fluorescence energy transfer)[15] have been used for the dual-emission ratiometric sensing of metal ions. However, they have rather limited utility as the former can only work with a select class of fluorophores and the latter demands a complex probe structure owing to the necessity of using two fluorophores. Given these limitations that restrict broader application of fluorescent probes for quantitative and accurate ratiometric analysis[16], there still exists a need for new molecular systems for the ratiometric detection of metal ions. We envisioned that such new sensing system should (1) work flexibly with different types of fluorophores, (2) function effectively within simple molecular architectures, and (3) be broadly applicable to various metal ions. These desirable functions would facilitate not only quantitative detection of various metal ions at a desired wavelength, but also identification of a specific metal ion with high accuracy. In this article, we report the development and applications of ratiometric fluorescent metal-ion probes that are based on contact interactions between the coordinated metal ion and the aromatic ring of the fluorophore (i.e., arene–metal-ion contact). Introduction of a semicyclic ligand to a tricyclic fluorophore provided a designer probe, which forms an arene–metal-ion van der Waals contact (AM-contact) with various d-block metal ions to cause large emission red-shifts. In particular, this molecular architecture enabled the ratiometric detection of 3d-block metal ions such as Zn(II) and Cu(II), which has not been achieved by AM-contact probes developed to date[17–22]. We also found that this probe design is broadly applicable to various tricyclic fluorophores, the emissions of which cover a broad wavelength range of over 400 nm. The cross-sensing ability of the AM-contact probes for the metal ions was successfully utilized to construct a multicolor fluorescent sensing system, which enabled us to distinguish eight metal ions by a one-pot single titration and principal component

analysis (PCA). The probe with the xanthene fluorophore was also applicable to the ratiometric detection of various metal ions under live-cell conditions. These findings demonstrate the broad utility of AM-contact sensing as a multicolor and ratiometric sensing platform for metal ions.

## Results

**Design of fluorescent probes**. We previously reported that a Type-I probe serves as the ratiometric fluorescent chemosensor for metal ions based on AM-contact sensing (Fig. 1)[17,18]. For example, probe **1**, bearing two 2,2′-dipicolylamine (i.e., $R = 2$-pyridyl) exhibited an emission red-shift upon binding to Cd(II), which was induced by the formation of a van der Waals contact between the C9 carbon of the fluorophore and the metal ion in a 1:1 binding complex. However, while Type-I probes fluorescently responded to 4d- and 5d-block metal ions such as Cd(II), Ag(I), and Hg(II), they did not show an emission shift with 3d-block metal ions such as Zn(II). This limitation was mainly ascribed to the unfavorable formation of 1:2 binding complexes with the rather small 3d-block metal ions at the ligand sites, which hindered the van der Waals contact with the fluorophore. To expand the applications of AM-contact sensing to 3d-block metal ions and demonstrate its broad utility as a sensing platform, we designed a Type-II probe (Fig. 1) possessing a single tetra-aza-cyclic ligand[23,24]. It was expected that the Type-II probe would preferentially form a 1:1 metal ion complex, resulting in a close van der Waals interaction. However, Type-II probes **2–4** were not able to fluorescently respond to most of the 3d-block metal ions such as Cr(II), Mn(II), Co(II), Ni(II), Zn(II), Ag(I), Cd(II), and Pb(II), with the exception of Cu(II) (Supplementary Fig. 1 and Supplementary Table 1). In the case of Cu(II), the fluorescence intensities decreased without an emission shift. We concluded that these undesirable results were primarily due to the rigidity of the conformationally constrained aza-cyclic ligands, which are connected to the anthracene ring at the 1,8-positions. We next designed a Type-III probe, which possessed a more flexible semicyclic ligand (Fig. 1). Probes **5–7**, each bearing a different semicyclic ligand, were synthesized according to standard methods, and their fluorescence response toward metal ions was evaluated under aqueous MeOH conditions (50 mM HEPES (pH 7.4)/MeOH = 1:1).

To our delight, probe **5** bearing a 1,9-bis(2′-pyridyl)-2,5,8-triazanonane (BPTN) ligand exhibited a clear emission red-shift ($\Delta F = 14$ nm) upon the addition of Zn(II) (Fig. 2a). The fluorescence change of **5** was consistent with a bathochromic absorption shift ($\Delta \text{Abs} = 6$ nm) (Fig. 2c), suggesting that Zn(II) coordination influences the electrophysical properties of the anthracene in the ground state. The fluorescence molar ratio plot clearly suggests that **5** forms a 1:1 binding complex with Zn(II) (Supplementary Fig. 2). The binding affinity of **5** to Zn(II) was $2.70 \times 10^6 \text{ M}^{-1}$ as evaluated by fluorescence titration (Supplementary Fig. 3 and Supplementary Table 2). Conversely, probes **6** and **7** showed negligible emission shifts upon the addition of Zn(II) (Supplementary Fig. 4). Probe **5** also displayed a large emission red-shift ($\Delta F = 65$ nm) and a bathochromic absorption shift ($\Delta \text{Abs} = 5$ nm) (Fig. 2b, d) upon complexation with Cd(II) ($K_a = 9.50 \times 10^6 \text{ M}^{-1}$) (Supplementary Fig. 3 and Supplementary Table 2), while **6** and **7** showed smaller emission shifts ($\Delta F = 6$ and 12.5 nm, respectively) compared to **5** (Supplementary Fig. 4).

**Ratiometric sensing with various fluorophores**. We next replaced the anthracene of **5** with other tricyclic fluorophores and evaluated the fluorescence sensing properties of this series of Type-III probes (Fig. 3). Probes **8–11** possess the fluorophores

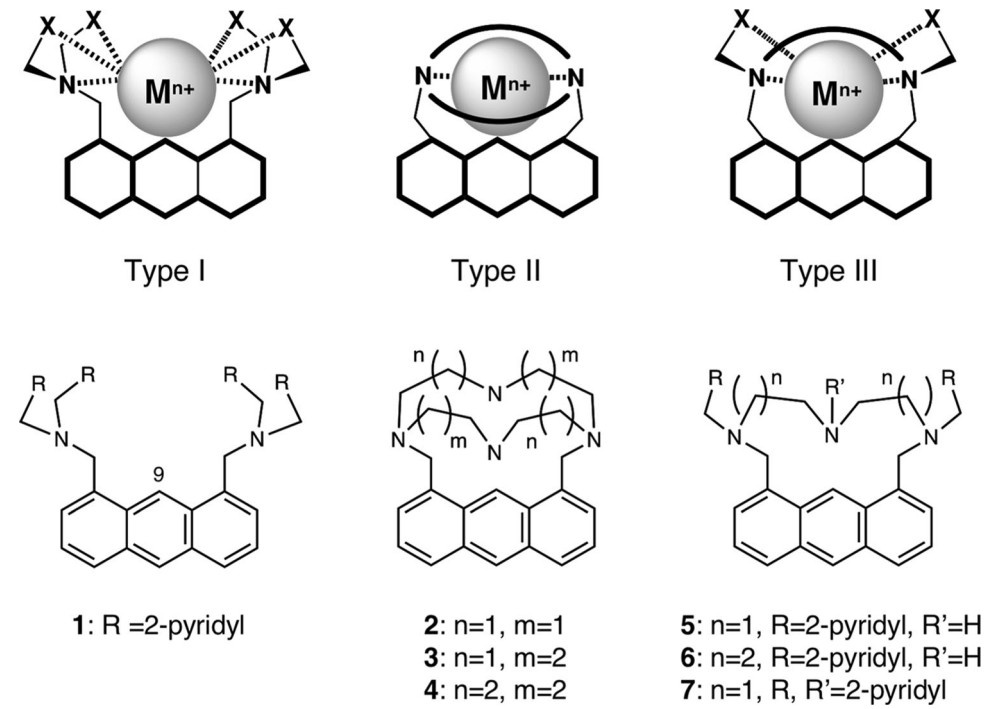

**Fig. 1 Design of AM-contact probe.** General designs (upper) and structures (lower) of tricyclic fluorescent probes for metal ion sensing based on arene–metal-ion contact (AM-contact).

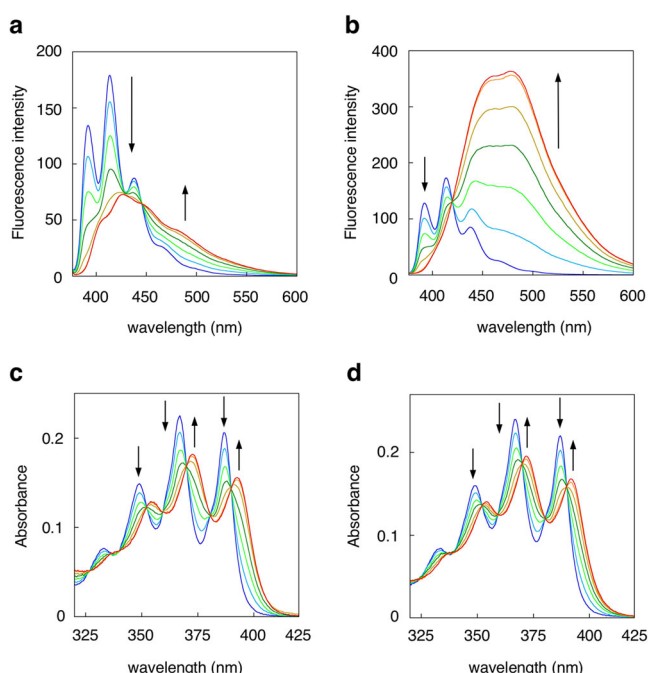

**Fig. 2 Fluorescence and absorption response of probe 5 toward metal ion.** Fluorescence and absorption spectral changes of probe **5** upon the addition of ZnCl$_2$ (**a**, **c**) or CdCl$_2$ (**b**, **d**). Measurement conditions: [**5**] = 25 μM, [ZnCl$_2$ or CdCl$_2$] = 0−30 μM, 50 mM HEPES (pH 7.4)/MeOH = 1:1, λ$_{ex}$ = 365 nm, 25 °C.

coumarin, xanthene, pyronine, and Si-pyronine, respectively. The emission wavelengths of **5** and **8–11** cover a wide range from 380 to 800 nm. In addition to the difference in the emission wavelengths, structural analysis of the fluorophores by density functional theory (DFT) suggested that the distance between the two methyl groups at their 1,8-positions (4,6-position in the case of

coumarin **8**) varies from 4.64 to 5.21 Å (Fig. 3). We expected that this structural difference also affects the sensing and binding properties of the Type-III probes to metal ions. The detailed synthetic procedures of the probes are described in the Supplementary information. The absorbance and fluorescence properties of the probes are summarized in Table 1.

Figure 4 shows the fluorescence spectra changes of **8–11** in titrations with Zn(II) and Cd(II) under aqueous MeOH conditions (50 mM HEPES (pH 7.4)/MeOH = 1:1). Xanthene **9** and pyronine **10** exhibited the largest emission red-shifts (ΔF = 27 and 25 nm, respectively) in the titration with Zn(II), while the wavelength shift of coumarin **8** was rather small (ΔF = 5 nm) (Table 2). Si-pyronine **11** exhibited a moderate emission red-shift (ΔF = 10 nm) in the near-infrared wavelength region (685 → 695 nm) upon coordination with Zn(II). Coordination to Cd(II) also induced a clear emission red-shift in the case of **9**, **10**, and **11** (ΔF = 27, 28, and 14 nm, respectively), while the wavelength shift of **8** was rather small (ΔF = 4 nm) as observed in the Zn(II) titration. It should be noted that all the probes showed bathochromic absorption shifts upon coordination with Zn(II) and Cd(II) (Supplementary Fig. 5 and Supplementary Table 3). Interestingly, the binding affinity for Zn(II) varied depending on the fluorophore (Supplementary Table 2). That is, anthracene **5** had the highest binding affinity ($K_a$ = 2.7 × 10$^6$ M$^{-1}$), which is ca. 4-fold and ca. 50-fold stronger than that of xanthene **9** ($K_a$ = 6.5 × 10$^5$ M$^{-1}$) and pyronine **10** ($K_a$ = 5.0 × 10$^4$ M$^{-1}$), respectively. This order of binding affinities was almost the same as that with Cd(II) (Supplementary Table 2). The weak binding affinities of pyronine **10** could be ascribed to the long distance between the two 1,8-methylene carbons on the pyronine fluorophore (Fig. 3), which may reduce the conformational flexibility of the BPTN ligand required for coordination.

We further investigated the fluorescence sensing properties of the probes toward other metal ions. All titrations were carried out under aqueous MeOH conditions (50 mM HEPES (pH 7.4)/ MeOH = 1:1) to avoid aggregation of the metal complex. Table 2 and Supplementary Fig. 6 summarize the maximum emission

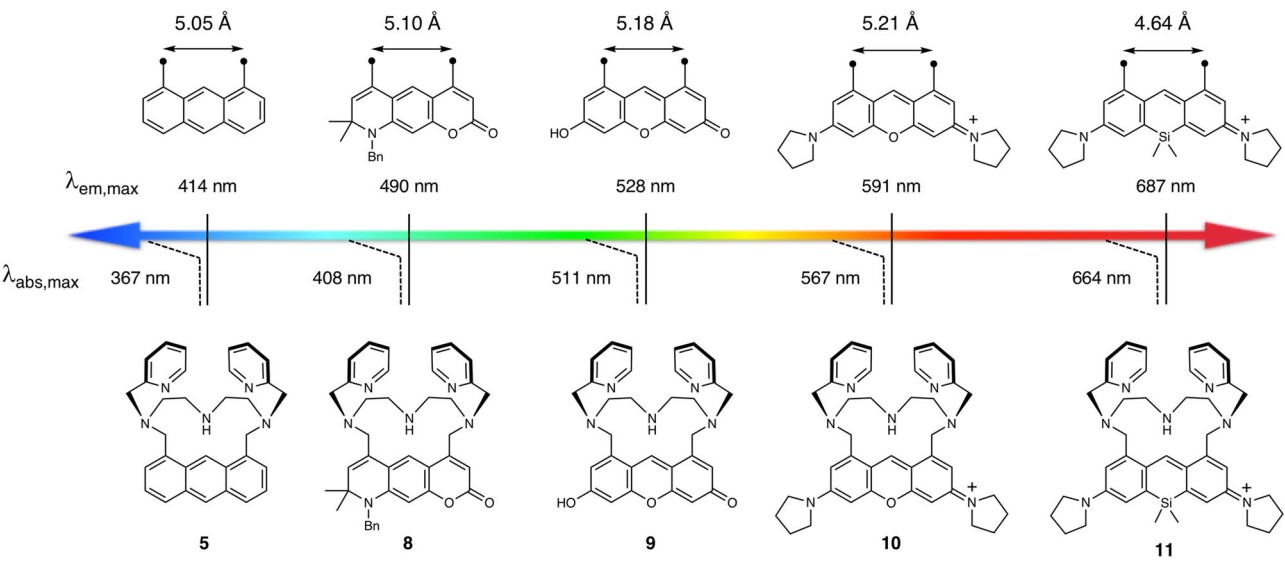

**Fig. 3 A series of AM-contact probes for metal ion sensing.** Structures and fluorescence spectroscopic properties of tricyclic AM-contact probes bearing a BPTN ligand.

**Table 1 Summary of optical properties of Type-III probes 5 and 8–11[a].**

|   | $\lambda_{abs,max}$ (nm) | $\varepsilon$ (cm$^{-1}$·M$^{-1}$) | $\lambda_{em,max}$ (nm) | $\Phi$ |
|---|---|---|---|---|
| **5** | 367 | 8500 | 414 | 0.03 |
| **8** | 408 | 19,500 | 490 | 0.50 |
| **9** | 511 | 66,000 | 528 | 0.24 |
| **10** | 567 | 138,000 | 591 | 0.53 |
| **11** | 664 | 138,500 | 687 | 0.30 |

[a]All measurements were conducted in 50 mM HEPES (pH 7.4)/MeOH = 1:1, 25 °C.

red-shifts ($\Delta F$) of **5** and **8–11** at the saturation point of the titration with the metal ions. All the probes except for coumarin **8** showed the clear emission red-shifts upon coordination with 4d- and 5d-block metal ions such as Ag(I), Cd(II), and Hg(II). In the titration with Cu(II), **9–11** also showed the clear emission red-shifts along with a large decrease in fluorescence (Supplementary Figs. 9–11). Conversely, anthracene **5** did not show an emission shift upon binding with Cu(II) (Supplementary Fig. 7), even though a substantial bathochromic shift ($\Delta Abs = 6$ nm) was observed in the absorption titration with Cu(II) (Supplementary Table 3). This observation can be explained by a very weak fluorescence emission of the copper complex of **5** as a result of the strong quenching effect of Cu(II). Similarly, **5** and **11** showed negligible emission red-shifts in the titrations with Ni(II) and Co(II), while their absorption wavelengths clearly shifted ($\Delta Abs = 6–15$ nm) (Supplementary Table 3). The metal ion titration was also conducted under aqueous buffer conditions (50 mM HEPES, pH 7.4) using the highly water-soluble probe **9**. Probe **9** exhibited a large emission red-shift upon binding with Cu(II), Zn(II), Ag(II), and Cd(II) (Supplementary Table 2). The small emission red-shifts of coumarin **8** toward all the metal ions tested (Supplementary Fig. 8) might be attributed to the formation of an ICT excited state, which cancels out the change in the photophysical property of the coumarin fluorophore induced by AM-contact. Another explanation for the small emission shifts of **8** might be the intrinsic low contact of the coumarin fluorophore because of its low electron density, as suggested by Hancock et al.[20]

**X-ray structures of the metal ion complexes**. The structures of the metal ion complexes of probe **5** were analyzed by X-ray crystallography (Fig. 5, Supplementary Fig. 12, and Supplementary Table 4). The zinc complex **5**-Zn(II) has a square pyramidal coordination geometry and the five nitrogen atoms are coordinated to the Zn(II) ion (Fig. 5a and Supplementary Data 1). The distance between Zn(II) and the adjacent C9 carbon atom is 2.96 Å, which is shorter than the sum of the van der Waals radii of Zn(II) (1.39 Å) and the aromatic carbon (1.77 Å) (Supplementary Table 5)[25]. Figure 5b is the cross-sectional view of the **5**-Zn(II) complex, which clearly shows the van der Waals contact between the Zn(II) ion and the C9 carbon. Interestingly, this close contact induces the protrusion of the C9 atom from the π plane of the anthracene ring by +4.9° (θ; dihedral angle of C10-C9a-C8a-C9), suggesting that the middle aromatic ring is bent away from Zn(II). van der Waals contact between the metal ion and fluorophore was also observed in **5**-Cd(II) (Fig. 5c and Supplementary Data 2) and **5**-Ag(I) (Fig. 5d and Supplementary Data 3). **5**-Cd(II) has a square pyramidal coordination geometry, and the distance between Cd(II) and the C9 carbon atom is 3.01 Å, which is shorter than the sum of the van der Waals radii of Cd(II) (1.58 Å) and the aromatic carbon (1.77 Å). **5**-Ag(I) has a trigonal pyramidal coordination geometry, in which one aliphatic nitrogen atom of the BPTN ligand does not coordinate to Ag(I). Although the distance between Ag(I) and the C9 carbon atom is rather long (3.28 Å), they still form a van der Waals contact owing to the large van der Waals radius of Ag(I) (1.72 Å). In contrast to **5**-Zn(II), the skew in the aromatic ring is negligible in **5**-Cd(II) and **5**-Ag(I) (θ = −0.4° and −0.1°, respectively). **5**-Cu(II) has a square planer coordination geometry, in which the distance between Cu(II) and the C9 carbon atom is 3.14 Å (Fig. 5e and Supplementary Data 4). Although this distance is only slightly shorter than the sum of the van der Waals radii of Cu(II) (1.40 Å) and the aromatic carbon (1.77 Å), the coordination of Cu(II) induced a substantial absorption shift ($\Delta Abs = 6$ nm) in **5** (Supplementary Table 3). It is also worth noting that the conformation of the BPTN ligand is sufficiently flexible to coordinate metal ions with different radii; the distance between the N3′ nitrogen and N9′ nitrogen of the BPTN unit varied from 4.22 to 4.54 Å depending on the metal ion (Supplementary Table 5), positioning the coordinated metal ions in close proximity to the anthracene ring and inducing the emission red-shift.

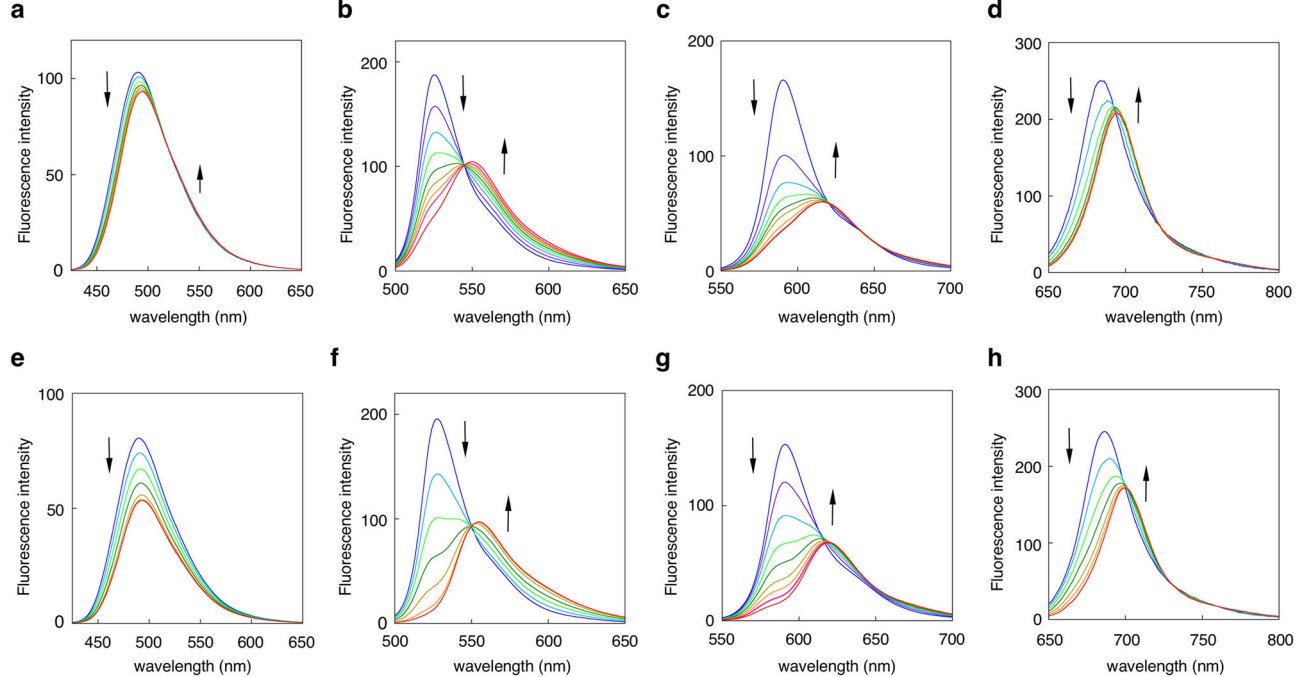

**Fig. 4 Fluorescence response of AM-contact probes toward metal ions.** Fluorescence spectral changes of probes **8**–**11** (from left to right) upon the addition of ZnCl₂ (**a**–**d**) or CdCl₂ (**e**–**h**). Measurement conditions: [probe] = 5 μM (**9**, **10**, **11**) or 10 μM (**8**), 50 mM HEPES (pH 7.4)/MeOH = 1:1, 25 °C.

---

**Table 2 Summary of fluorescence emission shifts (nm) of probes 5 and 8–11 upon the addition of various metal ions[a].**

| Probe | Cr³⁺ | Mn²⁺ | Co²⁺ | Ni²⁺ (1.63 Å)[b] | Cu²⁺ (1.40 Å)[b] | Zn²⁺ (1.39 Å)[b] | Ag⁺ (1.72 Å)[b] | Cd²⁺ (1.62 Å)[b] | Hg²⁺ (1.70 Å)[b] | Pb²⁺ (2.02 Å)[b] |
|---|---|---|---|---|---|---|---|---|---|---|
| **5** | _[c] (250)[d] | _[c] (250)[d] | <3 (250)[d] | <3 (250)[d] | <3 (50)[d] | 13 (30)[d] | 69 (250)[d] | 63 (30)[d] | 45 (250)[d] | 33 (250)[d] |
| **8** | _[c] (100)[d] | _[c] (100)[d] | <3 (100)[d] | <3 (100)[d] | <3 (50)[d] | 5 (50)[d] | <3 (20)[d] | 4 (50)[d] | <3 (15)[d] | <3 (20)[d] |
| **9** | _[c] (50)[d] | _[c] (50)[d] | <3 (50)[d] | <3 [3][e] (50)[d] | 25 [24][e] (10)[d] | 27 [25][e] (20)[d] | 8 [25][e] (50)[d] | 27 [25][e] (10)[d] | 34 (7)[d] | <3 (25)[d] |
| **10** | _[c] (250)[d] | _[c] (250)[d] | <3 (250)[d] | <3 (250)[d] | 30 (5)[d] | 25 (100)[d] | 15 (10)[d] | 28 (7)[d] | 35 (7)[d] | <3 (50)[d] |
| **11** | _[c] (50)[d] | _[c] (50)[d] | 8 (250)[d] | <3 (250)[d] | 13 (10)[d] | 10 (10)[d] | 4 (10)[d] | 14 (6)[d] | 16 (7)[d] | <3 (10)[d] |

[a]Fluorescence titration was conducted with a probe solution (25 μM of **5**, 10 μM of **8**, or 5 μM of **9**, **10**, **11**) in 50 mM HEPES buffer (pH = 7.4): MeOH = 1:1 at 25 °C. The excitation wavelengths of **5**, **9**, **10**, and **11** were 365, 410, 488, 578, and 674 nm, respectively. [b]The van der Waals radius of the metal ion. [c]Negligible fluorescence intensity change and emission shift were observed. [d]The concentration of metal ion (μM) at saturation point of emission change in titration. [e]The fluorescence emission shift (nm) observed under the aqueous buffer conditions (50 mM HEPES, pH 7.4).

---

**Computational analysis of AM-contact sensing**. To understand the theoretical mechanism of AM-contact sensing, we performed electronic-structure calculations for select probes and their zinc ion complexes. The low-lying excited states of xanthene probe **9** and its zinc complex **9**-Zn(II) were initially calculated with the time-dependent density functional theory (TDDFT) method (Table 3 and Supplementary Table 6). The $S_0 \rightarrow S_1$ transition energy of **9** was $\Delta E = 2.978$ eV in water, which is smaller than that of **9**-Zn(II) ($\Delta E = 2.733$ eV). This result reproduces the experimentally observed red-shift of the absorption band (2.426 eV (511 nm) → 2.380 eV (521 nm)) induced by Zn(II) ion coordination. Supplementary Table 6 summarizes the HOMO and LUMO energy levels of **9** and **9**-Zn(II). The HOMO-LUMO energy gap of **9** ($\Delta E = 2.978$ eV) decreases upon complexation with Zn(II) in water ($\Delta\Delta E = 0.245$ eV). A TDDFT calculation was also conducted by replacing the zinc ion of **9**-Zn(II) with a positive charge (PC) of $+1.0e$ (1PC) and $+2.0e$ (2PC) (Table 3). The $S_0 \rightarrow S_1$ transitions of **9** + PC(1.0) and **9** + 2PC(2.0) were

calculated to be 2.827 and 2.360 eV, respectively, under vacuum conditions, which are lower than that of **9** ($\Delta E = 3.015$ eV). The decrease of the $S_0 \rightarrow S_1$ transition energy was also induced by the coordination of other metal cations such as Na(I) and Ca(II) to **9** (Supplementary Table 7). These results all suggest that electrostatic interaction with positively charged Zn(II) influences the photophysical properties of **9**, leading to the coordination-induced emission red-shift in **9**-Zn(II). This prediction is consistent with the results previously reported for the Cd(II) and Ag (I) complexes of the Type-I probes[17].

In the case of the anthracene probe **5**, the $S_0 \rightarrow S_1$ transition energy in water was calculated to be 3.596 eV (Table 3). This value was reduced to 3.520 eV in the zinc complex **5**-Zn(II). This trend in the narrowing of the energy gap agrees with the experimentally observed red-shift of the absorption band (3.378 eV (367 nm) → 3.324 eV (373 nm)) induced by zinc coordination. In the calculation with a positive charge, the $S_0 \rightarrow S_1$ transition energy of **5**-PC(1.0) was calculated to be 3.576 eV under vacuum

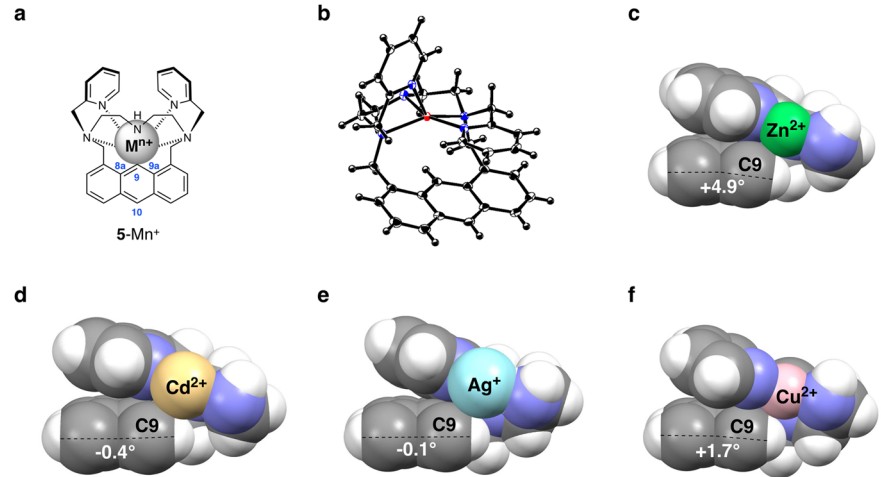

**Fig. 5 X-ray structures of the metal ion complexes of probe 5. a** General structure of the metal ion complex **5**-M$^{n+}$. **b** ORTEP diagram (50% probability ellipsoids) of **5**-Zn$^{2+}$ (C$_{32}$H$_{32}$N$_4$O$_8$Cl$_2$Zn). The perchlorate anions are omitted for clarity. **c–f** Cross-sectional views of the metal ion complexes of **5**-M$^{n+}$ (M$^{n+}$ = Zn$^{2+}$, Cd$^{2+}$, Ag$^+$, or Cu$^{2+}$). The angle shown in each complex is the dihedral angle of C10-C9a-C8a-C9.

**Table 3 Summary of the calculated S$_0$-S$_1$ excitation state of the probes with or without zinc ion/positive charge (PC) coordination.**

| Species | Optimized geometry[a] | Environment | S$_0$-S$_1$ excitation | | |
|---|---|---|---|---|---|
| | | | $\Delta E$ (eV) | $\lambda$ (nm) | $f$ |
| **9** | **9** | Water[b] | 2.978 | 416.3 | 0.808 |
| **9** + Zn(II) | **9** | Water[b] | 2.733 | 453.6 | 0.618 |
| **9** | **9** | Vacuum | 3.015 | 411.2 | 0.661 |
| **9** + PC(0.0) | **9**-Zn(II) | Vacuum | 3.003 | 412.9 | 0.570 |
| **9** + PC(1.0) | **9**-Zn(II) | Vacuum | 2.827 | 438.6 | 0.491 |
| **9** + PC(2.0) | **9**-Zn(II) | Vacuum | 2.360 | 525.3 | 0.098 |
| **5** | **5** | Water[b] | 3.596 | 344.8 | 0.163 |
| **5** + Zn(II) | **5** | Water[b] | 3.520 | 352.2 | 0.152 |
| **5** | **5** | Vacuum | 3.626 | 341.9 | 0.130 |
| **5** + PC(0.0) | **5**-Zn(II) | Vacuum | 3.549 | 349.3 | 0.113 |
| **5** + PC(1.0) | **5**-Zn(II) | Vacuum | 3.576 | 346.7 | 0.114 |
| **5** + PC(2.0) | **5**-Zn(II) | Vacuum | 3.555 | 348.8 | 0.112 |

conditions, which is lower than that of **5** ($\Delta E = 3.626$ eV) in a vacuum. This result is consistent with the emission red-shift observed for **5** coordinated to zinc. Interestingly, when the calculation was conducted for **5**-PC(0.0), which possesses the optimized geometry of **5**-Zn(II) with a bent anthracene ring, the S$_0 \rightarrow$ S$_1$ transition energy was calculated to be 3.549 eV. This value is smaller than that of **5**-PC(1.0) with the same geometry ($\Delta E = 3.576$ eV), contradicting the experimental results. This implies that the emission red-shift of **5** can be induced not only by the electronic perturbation of the positively charged zinc ion, but also by the deformation of the anthracene ring as observed in the X-ray crystal structure of **5**-Zn(II) (Fig. 5b). This hypothesis was also proposed for the Ag(I) complex of a Type-II probe in our previous paper[18].

**Differential sensing of metal ions**. Although the Type-III probes bearing a BPTN ligand can sense various metal ions, discrimination of individual metal ions is also of great importance in the fields of applied chemistry such as environmental monitoring, metal ion toxicology, and bioinorganic chemistry. In recent years, several types of fluorescence sensor arrays capable of distinguishing sets of metal ions have been reported[26–29]. These systems employ a series of cross-reactive sensors for metal ions, which produce a data set of

the fluorescence signal change. Applying statistical analysis to the data enables unambiguous identification of the metal ions. We expected that the probes using AM-contact sensing might also be applied to differential metal ion detection. In particular, we aimed to construct a one-pot multicolor fluorescence sensing system comprising a set of probes that fluoresce at different wavelengths. It was anticipated that this sensing system would allow us to avoid repeated measurements, which are usually necessary for sensor array detection.

As a proof-of-concept experiment, we employed the probes **5**, **9**, **10**, and **11** for multicolor metal ion sensing. A solution containing the set of probes (2 µM of **5**, **9**, and **10**, and 4 µM of **11**) displayed a complicated emission spectrum from 380 to 750 nm (Fig. 6a) when excited at 254 nm. The addition of Zn(II) (0–30 µM) induced a clear spectral change as a result of the dual-emission responses of each probe to provide a unique emission spectrum, which is distinct from the initial spectrum. Titration with other metal ions such as Co(II), Ni(II), Cu(II), Cd(II), Ag(I), Hg(II), and Pb(II) also provided unique emission spectral changes (Fig. 6b). Each spectral pattern was recorded at eight wavelengths to provide emission ratio values of $F_{450}/F_{414}$, $F_{550}/F_{525}$, $F_{615}/F_{589}$, and $F_{690}/F_{679}$, as well as emission intensity change $F/F_0$ values at 414, 525, 589, and 679 nm as a set of signal outputs. By analyzing

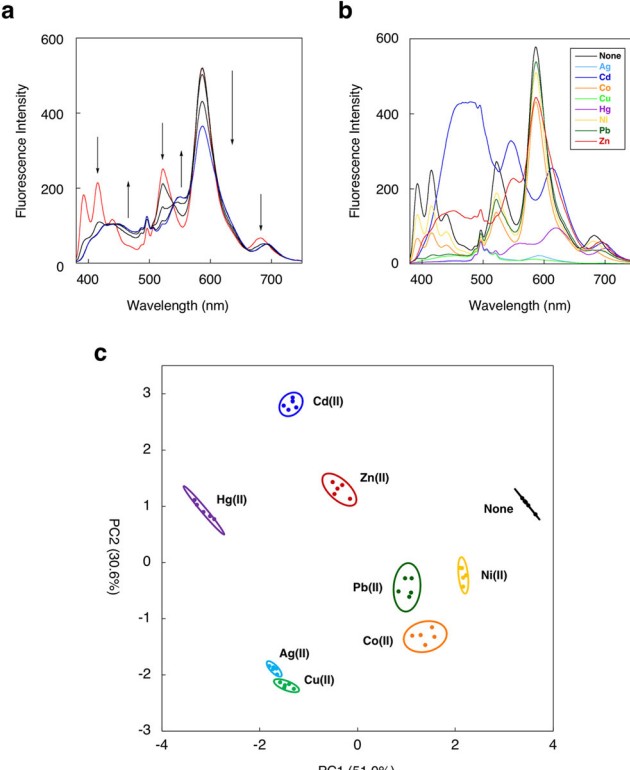

**Fig. 6 Multicolor fluorescent sensing of metal ions. a** Fluorescence spectral change of the set of probes **5**, **9**, **10**, and **11** upon the addition of ZnCl$_2$ (0−30 μM). **b** Fluorescence spectral change of the set of probes **5**, **9**, **10**, and **11** upon the addition of various metal ions (30 μM). **c** PCA plot of the two principal components (PC1 and PC2) for eight metal ions (30 μM). Here, 95% confidence circle is shown for each metal ion ($n = 5$ for each metal ion). Measurement conditions: [**5**, **9**, **10**] = 2 μM, [**11**] = 4 μM, 50 mM HEPES (pH 7.4)/MeOH = 1:1, 25 °C. $\lambda_{ex} = 254$ nm.

these data sets using PCA, we were able to differentiate the eight metal ions at 30 μM on a two-dimensional (2D) dispersion graph (Fig. 6c). Among the metal ions tested, Hg(II) lies furthest away from the control (none) sample on the 2D graph as Hg(II) coordination induced the large emission red-shifts in all the probes (Fig. 6b). Cd(II) also lies far away from the control on the 2D graph. This can be attributed mainly to the unique fluorescence emission change of the anthracene probe **5** induced by Cd(II) with a broad and large red-shifted spectrum. A similar but less pronounced spectral change was induced by Zn(II), enabling discrimination of Zn(II) from Cd(II). Cu(II) and Ag(II), both of which induced a large fluorescence decrease, were close to each other on the 2D graph. Nevertheless, the 95% confidence circles do not overlap on the 2D graph, affording reliable discrimination of the two metal ions. Among the observed variables used in PCA analysis, $F/F_0$ values at 414 and 525 nm largely contributed to both of the first and second principal components (PC1 and PC2), suggesting that the emission change of probe **5** contributed significantly to characterize the data set.

**Ratiometric imaging of metal ions in live cells**. We next set out to examine the utility of the xanthene-type probe **12** for ratiometric fluorescence imaging of metal ions in living cells (Fig. 7a). For this purpose, we designed compound **13**, which possesses the lipophilic O-*p*-acetoxy benzyl and N-ethoxyethyl groups. These substituents were introduced to increase the membrane permeability of **9**. We confirmed that probe **12** exhibits a clear

dual-emission change and a large increase in the fluorescence intensity ratio ($F_{550\,nm}/F_{520\,nm}$) upon the addition of ZnCl$_2$ in a neutral aqueous solution (50 mM HEPES, 100 mM NaCl, pH 7.4) (Fig. 7b, c). The binding affinity of **12** toward Zn(II) was evaluated to be $6.1 \times 10^6\,M^{-1}$ under the same aqueous conditions (Supplementary Fig. 13). When HeLa cells were treated with 5 μM of **12** in HBS buffer, bright fluorescence from the xanthene fluorophore was observed inside the cells (Fig. 7d). It was evident that non-fluorescent **13** penetrated the cells and was hydrolyzed to fluorescent **12** by intracellular esterases. Treatment of the cells with 5 μM of zinc chloride in the presence of pyrithione (100 μM) induced a clear fluorescence change inside the cells. Ratiometric fluorescence analysis revealed that the ratio value ($F_{540-630\,nm}/F_{500-530\,nm}$) gradually increased in a time-dependent manner. The imaging data using the spectral scan mode confirmed that the zinc ion induced the emission red-shift of **12** inside the cells (Supplementary Fig. 14). Live-cell imaging was further conducted with other metal ions such as Cu(II), Cd(II), and Hg(II) (Fig. 7e). The data showed that the fluorescence ratio value ($F_{540-630\,nm}/F_{500-530\,nm}$) changed inside the cells upon the addition of these metal ions (5 μM) in the presence of pyrithione (100 μM). These results are consistent with the emission red-shifts of **12** observed when titrating with these metal ions in an aqueous buffer (Supplementary Fig. 15). Although the poor metal ion selectivity of **12** did not allow selective detection of a specific metal ion, our results demonstrate that AM-contact sensing can work in living cells to enable ratiometric analysis of different metal ions.

## Discussion
Chelation-induced perturbation of ICT is the most widely used sensing mechanism in the development of ratiometric fluorescent probes for metal ions. In a probe that functions with this mechanism, the fluorophore must have a heteroatom (N, O, S, etc.) directly connected to or incorporated in its π-electron system, which undergoes a change in its electronic structure upon coordination with a metal ion, resulting in an emission wavelength shift. Unfortunately, this structural requirement limits the choice of fluorophores available for the ratiometric fluorescence sensing of metal ions. FRET is less frequently employed in the design of ratiometric fluorescence probes for metal ions because the FRET system requires two fluorophores within one molecule, making it difficult to design a probe with the proper function. The large molecular size of the FRET-type probe also inhibits its aqueous solubility and cell membrane permeability in live-cell imaging. In contrast to these existing systems, AM-contact sensing does not have such structural limitations and requirements because it utilizes the spatial proximity between the fluorophore and metal ion, allowing a variety of fluorophores to be employed in a simple molecular architecture. Indeed, we clearly demonstrated in this study that AM-contact sensing could operate with various tricyclic fluorophores bearing a BPTN ligand, enabling ratiometric detection of metal ions across a wide wavelength region. Of note, we successfully demonstrated that AM-contact sensing is operable with Si-pyronine, which emits a near-infrared fluorescence. This is a significant achievement in chemosensor research as the development of ratiometric near-infrared probes for metal ions is still a challenging task[30–32]. To our knowledge, this is the first example of Si-pyronine used in the ratiometric detection of metal ions.

Our previous results[17,18] and reports by Hancock[19–21] and Czarnik[22] suggested that the contact-induced emission shift was limited to rather large 4d- and 5d-block metal ions such as Cd(II) and Ag(I). In this paper, we demonstrated for the first time that AM-contact sensing also works with the smaller 3d-block

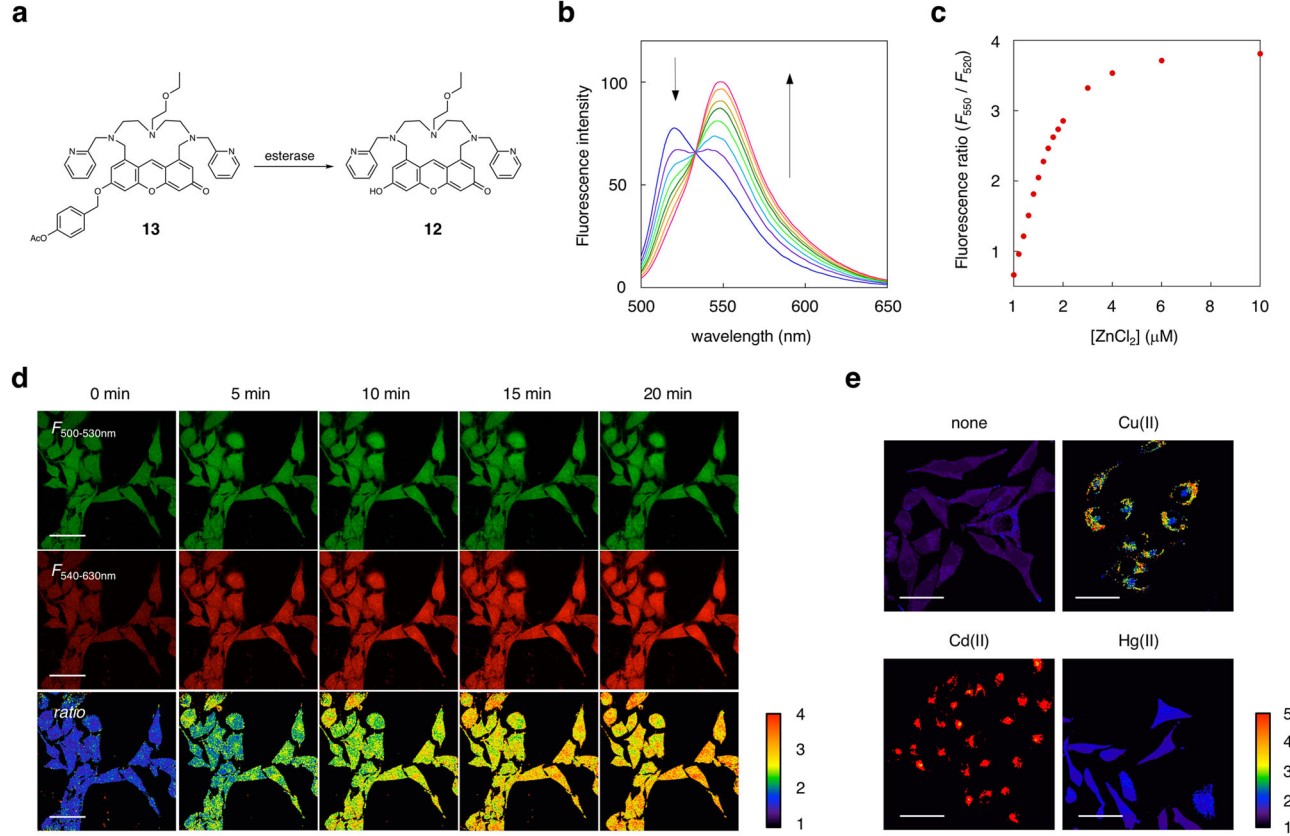

**Fig. 7 Ratiometric fluorescent detection of metal ions in living cells. a** Structures of probe **12** and **13**. **b** Fluorescence spectral change of **12** upon the addition of ZnCl$_2$. Measurement conditions: [**12**] = 1 µM, [ZnCl$_2$] = 0, 0.2, 0.4, 0.6, 0.8, 1.0, 1.4, 2 µM, 50 mM HEPES 100 mM NaCl, pH 7.4, 25 °C, $\lambda_{ex}$ = 488 nm. **c** Plot of the fluorescence intensity ratio ($F_{550 nm}$ / $F_{520 nm}$) of **12** upon the addition of ZnCl$_2$. **d** Time-lapse images of Zn(II) inside HeLa cells. The cells were incubated with **13** (5 µM) followed by the treatment with ZnCl$_2$ (5 µM) in the presence of pyrithione (100 µM). The ratio images were obtained from the imaging data acquired at two detection channels ($F_{540-630 nm}$ and $F_{500-530 nm}$). Scale bar: 50 µm. **e** Ratiometric detection of various metal ions in HeLa cells. The fluorescence images were acquired at 5 min after the treatment of the cells with each metal ion (5 µM) in the presence of pyrithione (100 µM). Scale bar: 50 µm.

metal ions such as Zn(II) and Cu(II). This was possible due to the molecular design of the Type-III probe, which employs the semicyclic BPTN as the metal ligand unit. Although the strong quenching effect of Co(II) and Ni(II) hampered the observation of an emission shift, their coordination to the probes **5** and **11** induced substantial absorption shifts (6–15 nm, Supplementary Table 3). These results bolster the possibility of ratiometric detection of these metal ions based on AM-contact. We expect that this might be achieved by employing other tricyclic fluorophores that are resistant to fluorescence quenching by Co(II) and Ni(II)[33–35]. X-ray crystallography of the metal ion complexes of **5** revealed that the BPTN ligand places a metal ion on the upper edge of the anthracene π-plane to facilitate van der Waals contact between them. Computational analysis suggested that the electrostatic interaction between the fluorophore and the adjacent metal ion largely contributes to the emission red-shift (Fig. 5). It is reasonable to assume that this electrostatic interaction can effectively work in a coordination complex wherein the metal ion is located close to the fluorophore, within their van der Walls radii. The computational analyses further suggested that the bend in the aromatic ring of the fluorophore, which was observed in the zinc complex of **5**, also contributes to the emission red-shift as a result of the change in the electronic structure of the fluorophore. We propose at this stage that these two structural perturbations (i.e., electrostatic interaction with the metal ion and deformation of the aromatic ring)

contribute to the emission shift in AM-contact sensing using Type-III probes.

We have demonstrated the utility of AM-contact sensing in two analytical applications: multicolor differential sensing of metal ions based on PCA analysis and ratiometric imaging of metal ions in living cells. Of note, we have conveniently distinguished eight metal ions by a one-pot single titration for differential sensing, which is difficult to achieve using the existing sensor array systems for differentiation of metal ions[26–29]. Both applications took full advantage of the unique properties of AM-contact sensing, including high compatibility with various fluorophores, broad applicability to different metal ions, and a clear ratiometric emission change. We expect that further refinement of the probe architecture, especially in the metal binding unit, will provide more sophisticated fluorescent probes with larger emission shifts and higher metal ion selectivities. We also envision that AM-contact sensing can be applicable to a broader range of metal ions, including alkaline and alkaline-earth metal ions, under various analytical settings[36]. Finally, we envisage that the utility of AM-contact sensing can be further extended to the dual-emission sensing of anions[37] and organic molecules[38] if their coordination to the metal ion can perturb the close contact between the fluorophore and the coordinated metal ion.

## Methods

**Metal ion titration**. Fluorescence spectra were recorded on a PerkinElmer LS-55 spectrofluorophotometer. Absorption spectra were measured using a Shimadzu

UV-2600 spectrophotometer. In a typical titration experiment, the probe solution (25 μM of **5**, 10 μM of **8**, 5 μM of **9**, **10**, **11**) in 50 mM HEPES buffer (pH = 7.4): MeOH = 1:1 was titrated with aqueous stock solution of metal ion in a quartz cell at 25 °C. The fluorescence and absorption spectra were measured 5 min after the addition of the metal ions at each titration point. In the fluorescence titration, probes **5**, **8**, **9**, **10**, and **11** were excited at 365, 410, 488, 578, and 674 nm, respectively. The plot of the fluorescence intensity at the maximum emission wavelength was analyzed by nonlinear least-square curve fitting to obtain the binding constant ($K_a$, $M^{-1}$).

**Determination of fluorescence quantum yield (Φ)**. Fluorescence quantum yields (Φ) of the probes were measured by comparison of the integrated area of the corrected emission spectrum of the probes in 50 mM HEPES buffer (pH 7.4) with those of the quantum yield standards: quinine sulfate (Φ = 0.55) for **5** and **8**, fluorescein (Φ = 0.95 in 1 N NaOH) for **9**, Rhodamine 6G (Φ = 0.94 in EtOH) for **10**, and NileBlue (Φ = 0.27 in EtOH) for **11**[39,40]. To minimize the re-absorption effect, fluorescence spectra were measured at the concentration where absorbance at the excitation wavelength was below 0.02. For the determination of Φ of **9**, **10**, and **11**, refractive indices of $H_2O$ ($n$ = 1.333) and EtOH ($n$ = 1.361) were used to correct the fluorescence intensities.

**Multicolor fluorescence sensing of metal ions**. Metal ion titration was carried out in an aqueous-methanol solvent (50 mM HEPES buffer (pH = 7.4):methanol = 1:1) containing the set of probes 2 μM of **5**, **9**, **10** and 4 μM of **11**. After the addition of the metal ions, the fluorescence spectra were recorded on a PerkinElmer LS-55 spectrofluorophotometer at 25 °C ($λ_{ex}$ = 254 nm). PCA was performed with Microsoft Excel 2011 using the data set of the emission intensity change ($F/F_0$ at 414, 525, 589, and 679 nm) and the emission ratios ($F_{450}/F_{414}$, $F_{550}/F_{525}$, $F_{615}/F_{589}$, and $F_{690}/F_{679}$).

**X-ray crystallography**. All the metal ion complexes of **5** were crystallized from MeOH/$CH_3CN$ = 3:1 at 30 °C. The X-ray data were collected on a Bruker AXS APEX II diffractometer with graphite monochromated MoKa radiation (λ = 0.71069 Å). The structures were solved by the direct method and refined anisotropically for non-hydrogen atoms by full-matrix least-squares calculation. The crystallographic calculations were performed by using the CrystalStructure Ver. 4.2 (Rigaku Corporation).

**Theoretical computational analysis**. Density functional theory (DFT) was applied to investigate the electronic origin of the spectral shift induced in the metal complexes. In the calculation, the CAM-B3LYP functional was applied to describe the exchange-correlation term. This calculation was carried out for molecules in a vacuum with the default computational parameters in Gaussian09. In order to calculate the excited states, the TDDFT method was applied to obtain the lower 20 states. The Kohn–Sham orbitals were described with the Gaussian basis sets. For all the elements, we used the 6-31G(d,p) and the 6-31+G(d,p) basis sets for optimization. The solvent effect was taken into account by using the polarized continuum model for water as the solvent.

**Cell culture**. HeLa cells were cultured in high-glucose Dulbecco's Modified Eagle Medium (DMEM, 4.5 g of glucose per litter, Sigma-Aldrich) supplemented with 10% fetal bovine serum (FBS), penicillin (100 unit $mL^{-1}$) and streptomycin (100 μg $mL^{-1}$) under a humidified atmosphere of 5% $CO_2$ in air. Subculture was performed every 3–4 days from subconfluent (~80%) cultures using a trypsin-EDTA solution.

**Ratiometric fluorescence imaging of metal ions in living cells**. HeLa cells (1 × $10^5$) were cultured on 3.5 cm glass-bottomed dish (Iwaki) for 2 days at 37 °C in a $CO_2$ incubator. The cells were washed with HBS (20 mM HEPES, 107 mM NaCl, 6 mM KCl, 1.2 mM $MgSO_4$, 2 mM $CaCl_2$, 11.5 mM glucose, adjusted to pH 7.4 with NaOH) twice and pre-treated with probe **13** (5 μM) in HBS(+) for 20 min at 37 °C. After washing with HBS(+), the cells were treated with HBS containing pyrithione (100 μM) and each metal ion (5 μM) for 30 min at 37 °C and subjected to fluorescence imaging using a confocal microscope (TCS SP8, Leica Microsystems) equipped with a HyD detector ($λ_{ex}$ = 488 nm).

**Synthetic procedures**. Detailed synthetic procedures for the probes are described in the Supplementary information.

**Reporting summary**. Further information on research design is available in the Nature Research Reporting Summary linked to this article.

## Data availability

The authors declare that all data supporting the findings of this study are available within the article and Supplementary information files, and from the corresponding author on request. The X-ray crystallographic coordinates for structures of **5**-Zn(II), **5**-Cd(II), **5**-Ag (I), and **5**-Cu(II) reported in this article have been deposited at the Cambridge Crystallographic Data Centre (CCDC), under deposition numbers CCDC 1993118–1993121, respectively. These data can be obtained free of charge from The Cambridge Crystallographic Data Centre via www.ccdc.cam.ac.uk/data_request/cif.

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

## Acknowledgements

This work was supported by the Grant-in-Aid for Scientific Research on Innovative Areas "Chemistry for Multimolecular Crowding Biosystems" (JSPS KAKENHI Grant No. JP17H06349) and the Grant-in-Aid for Scientific Research B (JSPS KAKENHI Grant No. JP20H02861 to A.O.). A.O. acknowledges Naito Science Foundation and Toray Science Foundation for their financial supports. S.U. acknowledges Grant-in-Aid for Young Scientists B (JSPS KAKENHI Grant No. JP17K14518) for its financial support.

## Author contributions

A.O. designed the experiments. A.K., Y.T., I.T., S.U., R.K. and A.Y. performed the experiments. U.K. and M.S. performed the computational theoretical analyses. A.O., J.W. and M.S. analyzed and discussed the data. The manuscript was written by A.O., S. U., J.W. and M.S.

## Competing interests

The authors declare no competing interests.
