## [Peer Review File. · Communications Chemistry]

Reviewers' comments:

Reviewer #1 (Remarks to the Author):

In this paper, the authors reported a series of new fluorescence sensor that can detect transition metal ions. The design is based on several kinds of tricyclic fluorophores that are connected to a semi-cyclic metal binding site. The metal binding efficiently alters the natures of the fluorophore through the arene-metal-ion contact as the authors claimed. The paper well described the background of this research in the introduction and also rationally stated the optimization of the ligand structures from the viewpoint of the metal binding sites and fluorophores. In the results and discussion sections, the authors comprehensively summarized the fluorescence responses, plausible binding structures, ratiometric analysis, and bioimaging applications. The paper is generally well-organized and could be suitable for publication in Communications Chemistry.

Detailed comments:

- In Table 2, the authors summarized the fluorescence data according to the emission shifts. These are actually dependent on the ratios of the guest-bound species to the guest-free species, which depend on the binding strengths, the equivalents of the guests, concentrations, measurement conditions, etc. Therefore, the Table 2 should include other important data such as binding constants and added amounts of the guests, in addition to footnotes containing the detailed measurement conditions.
- In the discussion of the crystal structures, the authors described the bending of the C-H bond from the pi plane. The authors should state the method of the X-ray analysis refinements of the C-H hydrogen atoms, since most of the recent X-ray crystal structural analyses use the riding models to determine the coordinates of hydrogen atoms, which is not suitable for the geometries regarding hydrogen atoms.
- NH hydrogen is needed in the structural diagram in Fig 5.

Reviewer #2 (Remarks to the Author):

This manuscript describes the preparation of a new set of metal-responsive probes based on arene-metal-ion contact. The probes show a colour change upon addition of some metal ions, and the authors demonstrate the potential use in array-based sensing and in microscopy studies. A large amount of work (spanning synthesis, crystallography, DFT calculations, biological studies and photophysical characterisation) are described here. However, before publication is appropriate, the authors need to clarify the aims of the work, and address some other issues as described below.

1. The authors fail to make clear in the introduction what the aim of this work is, and this therefore diminishes any results that they subsequently report. In particular, it appears that the aim is for non-selective sensing (since a probe that responds only to Cu(II) is characterised as "unwanted"), but why is this? Furthermore, there is a significant challenge in distinguishing Cd(II) from Zn(II) by fluorescent sensing, which the authors have achieved here, yet they do not acknowledge the value of this result. The introduction should clearly state the aims of this work (in the context of current needs in the field), and the rest of the manuscript show how these aims are achieved. This should include the desire for a cross-reactive probe, and what other properties of the probe are desired.
2. The introduction mentions a number of common fluorescent sensing methods, but there is no mention of systems where ICT is perturbed – this should be included
3. Page 2, line 65 – "facilitate not only quantitatively detect various" should be "facilitate not only quantitative detection of various".
4. The introduction suggests that arene-metal-ion contact has not previously been demonstrated as a mechanism for fluorescent metal sensing, but this is not the case, and indeed a number of other examples are provided in the conclusion. These references should be moved to the introduction, at the end of page 2.
5. The drawbacks of the Type-I probes is cited as a failure to respond to 3d-block metal ions, but is this really a disadvantage? A fluorescent sensor that can differentiate 4d and 5d metal ions from 3d metal ions is surely very useful.
6. Table 2 heading should specify the units (presumably nm)

7. Table 3 heading should be amended to indicate that these are calculated values
8. It is not appropriate to present PC analysis of data where only 3 replicates have been performed. The experiments shown in figure 6c should be repeated with at least 5 replicates.
9. Figure 6 caption – “principle” should be “principal” (also Figure S13). Also for Figure 6b, the authors should specify which “set of probes” is being mentioned.
10. The authors should provide some more discussion on the PCA shown in Figure 6c – what can the principal components tell us which probe contributes most to the diversity in the data?
11. It is not clear how the data was generated for Figure S13. The principal components are different from those derived in figure 6c, and yet only 1 replicate for each concentration has been carried out. It seems that the authors have omitted key data here, and they should provide the original PC analysis that was used to derive Figure S13.
12. It is fine to show the cell studies, but in reality such a probe would not be very useful in cell studies as it is so poorly selective. This should be acknowledged in the text.
13. Figure 7 should include the cuvette fluorescence response of the probe to Cu, Cd and Hg.
14. It would be ideal if the authors could carry out a spectral scan of cells, rather than just taking the two separate windows and measuring the ratio, but I realise that this might not be possible with current instrumentation.
15. Methods – Metal ion titration – multiple scans were taken, so the first sentence should be “Fluorescence spectra were recorded...”. Same for the 2nd sentence. The 3rd sentence needs “a” between “In” and “typical”
16. References should be provided for the quantum yields of the standards. Also, the method of measuring quantum yields should be explained more fully.
17. It would help to have a comment at the end of the methods section indicating that synthetic methods are provided in the supplementary information.

Reviewer #3 (Remarks to the Author):

The authors describe a series of fluorescent triaryl scaffolds with a bis pyridyl azacrown ligand (BPTN) as sensors for metals. The work is well executed and will certainly be of interest to the large number of groups working in the field of imaging and sensing. The sensing aspect is demonstrated using a PCA approach where identification of a number of metals is accomplished. A small set of metals were also successfully indicated inside cells.

In terms of novelty Bis methyleneamino anthracene and related triaryls have been used as scaffolds before – including the excellent ZnDPA work of the author and also the 2015 ChemBioChem manuscript (also by the author) where arene-metal contact was demonstrated using a range of xanthone scaffolds (similar to those in the current study). Yoon (and the current author) has also used this platform and (Inorg Chem 2014) appended crowns to for the purpose of metal sensing. As such elements of the current approach have been previously demonstrated.

Nevertheless the previous work was limited to very specific targets and the new generation of probes with the BPTN ligand described here (demonstrated on a range of fluorescent scaffolds) illustrates that the arene metal concept can be used in a more general sense.

While the probes are a definite improvement on prior systems they are not completely “universal” and the range of metals successfully indicated (Cu, Zn, Ag, Cd and Hg) is not extensive. It is only by using a PCA approach that some of the metals that give only a subtle spectroscopic change (such as Ni) can also be distinguished. Only four metals can be quantified.

Though the work is well executed and is certainly publishable I think this manuscript falls just short in terms of the very high novelty criteria required for this particular journal.

Additional: The ESI is very well organised but the actual NMR spectra of the new probes were not provided in the submission I reviewed.

Reply to the comments of the reviewer 1

Comment 1: In Table 2, the authors summarized the fluorescence data according to the emission shifts. These are actually dependent on the ratios of the guest-bound species to the guest-free species, which depend on the binding strengths, the equivalents of the guests, concentrations, measurement conditions, etc. Therefore, the Table 2 should include other important data such as binding constants and added amounts of the guests, in addition to footnotes containing the detailed measurement conditions.

Reply: In accordance with the comment, we modified Table 2 to include the concentrations of the metal ions at the saturation point of the emission change in the titration. We also added the detailed fluorescence titration conditions in the Table footnote. The corresponding fluorescence spectral changes in the titration were shown in Figure 2 and 4, and the supplementary information (Figure S7–S11). The binding constants of the probes for the representative metal ions were described separately in Table S2.

Comment 2: In the discussion of the crystal structures, the authors described the bending of the C-H bond from the pi plane. The authors should state the method of the X-ray analysis refinements of the C-H hydrogen atoms, since most of the recent X-ray crystal structural analyses use the riding models to determine the coordinates of hydrogen atoms, which is not suitable for the geometries regarding hydrogen atoms.

Reply: We appreciate this helpful comment. In our X-ray analyses, we used the standard rigid model to predict the position of the hydrogen atoms. We thus changed the method of geometrical analysis and reassessed the conformation of the aromatic ring based on the geometric data of the aromatic carbons. In the new analysis, we measured the dihedral angles of C10-C9a-C8a-C9 of the metal complexes and compared the values to each other. The data revealed that the dihedral angle ($\theta = +4.9^\circ$) of the zinc complex is larger than those of other metal complexes ($\theta = -0.4 - +1.7^\circ$), suggesting that the anthracene ring of the zinc complex forms a bent structure. In the revised manuscript, we modified Figure 5 to explain these results and modified the main text (page 8) as follows: Interestingly, this close contact induces the protrusion of the C9 atom from the π plane of the anthracene ring by $+4.9^\circ$ (θ ; dihedral angle of C10-C9a-C8a-C9), suggesting that the middle aromatic ring is bent away from Zn(II).

Comment 3: NH hydrogen is needed in the structural diagram in Fig 5.

Reply: We corrected the structure of the complex accordingly.

Reply to the comments of the reviewer 2

Comment 1: The authors fail to make clear in the introduction what the aim of this work is, and this therefore diminishes any results that they subsequently report. In particular, it appears that the aim is for non-selective sensing (since a probe that responds only to Cu(II) is characterised as “unwanted”), but why is this? Furthermore, there is a significant challenge in distinguishing Cd(II) from Zn(II) by fluorescent sensing, which the authors have achieved here, yet they do not acknowledge the value of this result. The introduction should clearly state the aims of this work (in the context of current needs in the field), and the rest of the manuscript show how these aims are achieved. This should include the desire for a cross-reactive probe, and what other properties of the probe are desired.

Reply: We appreciate this valuable comment. In accordance with the comment, we revised the introduction to more clearly state the aim of this work. The primary aim of the current work is to demonstrate the broad utility of AM-contact as a ratiometric sensing platform for metal ions including the 3d-block Zn(II) and Cu(II). We believe that our experimental results, which are collected based on the design of a new type of AM-contact probe, clearly demonstrate the achievement of this goal. To more clearly state this point, the introduction was modified as follows: In this article, we report the development and applications of ratiometric fluorescent metal ion probes that are based on contact interactions between the coordinated metal ion and the aromatic ring of the fluorophore (*i.e.*, arene–metal-ion contact). Introduction of a semicyclic ligand to a tricyclic fluorophore provided a designer probe, which forms an arene–metal-ion van der Waals contact (AM-contact) with various d-block metal ions to cause large emission red-shifts. In particular, this molecular architecture enabled the ratiometric detection of 3d-block metal ions such as Zn(II) and Cu(II), which has not been achieved by AM-contact probes developed to date.¹⁷⁻²² We also found that this probe design is broadly applicable to various tricyclic fluorophores, the emissions of which cover a broad wavelength range of over 400 nm. The cross-sensing ability of the AM-contact probes for the metal ions was successfully utilized to construct a multicolor fluorescent sensing system, which enabled us to distinguish 8 metal ions by a one-pot single titration and principal component analysis (PCA). The probe with the xanthene fluorophore was also applicable to the ratiometric detection of various metal ions under live-cell conditions. These findings demonstrate the broad utility of AM-contact sensing as a multicolor and ratiometric sensing platform for metal ions.

In addition, we also changed the manuscript title to emphasize the achievements in this work as follows: “A Multicolor and Ratiometric Fluorescent Sensing Platform for Metal Ions Based on Arene–Metal-Ion Contact”

Comment 2: The introduction mentions a number of common fluorescent sensing methods, but there is no mention of systems where ICT is perturbed – this should be included.

Reply: In the original manuscript, we used the term “PCT (photoinduced charge transfer)” to explain the sensing mechanism involving perturbation of the intramolecular charge transfer (ICT) state of the fluorophore in a photo-excited state. However, it seems that ICT is more commonly used to explain this sensing mechanism in relevant papers recently published. We thus replaced the term “PCT (photoinduced charge transfer)” with “ICT (intramolecular charge transfer)” in the revised manuscript.

Comment 3: Page 2, line 65 – “facilitate not only quantitatively detect various” should be “facilitate not only quantitative detection of various”.

Reply: We corrected the sentence pointed out by the reviewer accordingly.

Comment 4: The introduction suggests that arene-metal-ion contact has not previously been demonstrated as a mechanism for fluorescent metal sensing, but this is not the case, and indeed a number of other examples are provided in the conclusion. These references should be moved to the introduction, at the end of page 2.

Reply: We appreciate this helpful comment. To avoid the misunderstanding pointed out by the reviewer, we moved all of the related references to the introduction.

Comment 5: The drawbacks of the Type-I probes is cited as a failure to respond to 3d-block metal ions, but is this really a disadvantage? A fluorescent sensor that can differentiate 4d and 5d metal ions from 3d metal ions is surely very useful.

Reply: We agree with this opinion that the metal ion selectivity is surely an important point when assessing the utility of a fluorescent probe. However, as claimed in the introduction, the primary aim of the present study is to demonstrate the broad utility of AM-contact as a general sensing platform. This paper thus focused on the increase of metal ions species (particularly 3d-block metal ions) detectable by AM- contact sensing. From this point of the view, the sensing ability of the type-I probe being limited to 4d- and 5-block metals is considered to be a disadvantage. To more clearly explain this point, we modified the sentence in the introduction (page 4) as follows: To expand the applications of AM-contact sensing to 3d-block metal ions

and demonstrate its broad utility as a sensing platform, we designed a Type-II probe (Figure 1) possessing a single tetra-aza cyclic ligand.

Comment 6: Table 2 heading should specify the units (presumably nm).

Reply: We added the unit (nm) in the heading of Table 2 accordingly.

Comment 7: Table 3 heading should be amended to indicate that these are calculated values

Reply: We changed the heading of Table 3 accordingly.

Comment 8: It is not appropriate to present PC analysis of data where only 3 replicates have been performed. The experiments shown in figure 6c should be repeated with at least 5 replicates.

Reply: In accordance with the comment, we conducted the PC analysis with 5 replicates. The revised data shown in Figure 6c suggests that our probe can differentiate the 8 metal ions with sufficient reliability.

Comment 9: Figure 6 caption – “principle” should be “principal” (also Figure S13). Also for Figure 6b, the authors should specify which “set of probes” is being mentioned.

Reply: We corrected the places pointed out by the reviewer.

Comment 10: The authors should provide some more discussion on the PCA shown in Figure 6c – what can the principal components tell us which probe contributes most to the diversity in the data?

Reply: We appreciate this helpful comment. In accordance with the suggestion, we more carefully analyzed the data and added the following sentences in the revised manuscript (page 13): Among the metal ions tested, Hg(II) lies furthest away from the control (none) sample on the 2D graph as Hg(II) coordination induced the large emission red shifts in all the probes (Figure 6b). Cd(II) also lies far away from the control on the 2D graph. This can be attributed mainly to the unique fluorescence emission change of the anthracene probe **5** induced by Cd(II)

with a broad and large red-shifted spectrum. A similar but less pronounced spectral change was induced by Zn(II), enabling discrimination of Zn(II) from Cd(II). Cu(II) and Ag(II), both of which induced a large fluorescence decrease, were close to each other on the 2D graph. Nevertheless, the 95% confidence circles do not overlap on the 2D graph, affording reliable discrimination of the two metal ions. Among the observed variables used in PCA analysis, F/F_0 values at 414 and 525 nm largely contributed to both of the first and second principal components (PC1 and PC2), suggesting that the emission change of probe **5** contributed significantly to characterize the data set.

Comment 11: It is not clear how the data was generated for Figure S13. The principal components are different from those derived in figure 6c, and yet only 1 replicate for each concentration has been carried out. It seems that the authors have omitted key data here, and they should provide the original PC analysis that was used to derive Figure S13.

Reply: In accordance with this comment, we reconducted this concentration-dependent titration repeatedly (more than five times) to check the reproducibility of the data. Unfortunately, we were not able to obtain reliable data that showed clear discrimination of the four metal ions (Zn, Cu, Cd, Hg) in the concentration range. Since this result cannot support our claim, we decided to omit this result (Figure S13 in the original manuscript) from the revised manuscript.

For additional information, we conducted this concentration-dependent titration according to the reports by Chang et al. (Anal. Chem., 2014, 86, 8763) and Kool et al. (JACS, 2014, 136, 14576). We conducted the usual PCA analysis with the whole titration data (0 to 30 μ M) of the four metal ions.

Comment 12: It is fine to show the cell studies, but in reality such a probe would not be very useful in cell studies as it is so poorly selective. This should be acknowledged in the text.

Reply: In accordance with the comment, we modified the main text (page 14) to acknowledge the poor metal ion selectivity of the probe as follows: Although the poor metal ion selectivity of **13** did not allow selective detection of a specific metal ion, our results demonstrate that AM-contact sensing can work in living cells to enable ratiometric analysis of different metal ions.

Comment 13: Figure 7 should include the cuvette fluorescence response of the probe to Cu, Cd and Hg.

Reply: In accordance with the comment, we added the results of the fluorescence titration with Cu, Cd, and Hg in the Supplementary Information (Figure S15).

Comment 14: It would be ideal if the authors could carry out a spectral scan of cells, rather than just taking the two separate windows and measuring the ratio, but I realise that this might not be possible with current instrumentation.

Reply: To address this point, we re-synthesized probe **14** and conducted the live cell imaging using the spectral scan mode. The results showed that the fluorescence emission of the probe red-shifted with zinc inside the cells. This result supports the ratiometric imaging data shown in Figure 7d. The data of the spectral change was added in the Supplementary Information (Figure S14) of the revise manuscript.

Comment 15: Methods – Metal ion titration – multiple scans were taken, so the first sentence should be “Fluorescence spectra were recorded...”. Same for the 2nd sentence. The 3rd sentence needs “a” between “In” and “typical”.

Reply: We corrected the sentences accordingly.

Comment 16: References should be provided for the quantum yields of the standards. Also, the method of measuring quantum yields should be explained more fully.

Reply: We added the references 40 and 41 for the original data of the fluorescence quantum yields of the standards. We also described the detailed procedure for determining the quantum yield in the revised manuscript (page 17).

Comment 17: It would help to have a comment at the end of the methods section indicating that synthetic methods are provided in the supplementary information.

Reply: We added the following sentence in the Methods section: Detailed Synthetic procedure for the probes are described in Supplementary Information.

Reply to the comments of the reviewer 3

Comment 1: The authors describe a series of fluorescent triaryl scaffolds with a bis pyridyl azacrown ligand (BPTN) as sensors for metals. The work is well executed and will certainly be of interest to the large number of groups working in the field of imaging and sensing. The sensing aspect is demonstrated using a PCA approach where identification of a number of metals is accomplished. A small set of metals were also successfully indicated inside cells.

In terms of novelty Bis methyleneamino anthracene and related triaryls have been used as scaffolds before – including the excellent ZnDPA work of the author and also the 2015 ChemBioChem manuscript (also by the author) where arene-metal contact was demonstrated using a range of xanthone scaffolds (similar to those in the current study). Yoon (and the current author) has also used this platform and (Inorg Chem 2014) appended crowns to for the purpose of metal sensing. As such elements of the current approach have been previously demonstrated.

Nevertheless the previous work was limited to very specific targets and the new generation of probes with the BPTN ligand described here (demonstrated on a range of fluorescent scaffolds) illustrates that the arene metal concept can be used in a more general sense.

While the probes are a definite improvement on prior systems they are not completely “universal” and the range of metals successfully indicated (Cu, Zn, Ag, Cd and Hg) is not extensive. It is only by using a PCA approach that some of the metals that give only a subtle spectroscopic change (such as Ni) can also be distinguished. Only four metals can be quantified.

Though the work is well executed and is certainly publishable I think this manuscript falls just short in terms of the very high novelty criteria required for this particular journal.

Reply: We appreciate this useful comment, which suggests the pros and cons of the original version of our manuscript. In the revision, we modified the main text (particularly in the introduction and discussion sections) to more clearly state the significance and novelty of our work.

In the current work, we, for the first time, revealed that AM-contact sensing effectively works for dual-emission sensing not only for a limited number of 4d- and 5d-block metal ions, but also for 3d-block metal ions such as Zn and Cu. This should be regarded as significant progress in the fluorescence sensing field because no one had noticed that the molecular design of the probe is the key to achieving AM-contact sensing of 3d-block metal ions such as Zn(II) and Cu(II). Although other 3d-block metals such as Co(II) and Ni(II) did not induce a clear emission shift with our probes (Table 2) due to their strong fluorescence quenching effects, we demonstrated that the coordination of these metal ions induced substantial absorption red-shifts (6-15 nm) in the probes **5** and **11** (Table S3). These results suggest the possibility of ratiometric detection of

these metal ions based on AM-contact. We expect that this can be achieved by employing other tricyclic fluorophores, which are resistant to the quenching effect of these metal ions. This point is discussed in the discussion section of the revised manuscript (page 15) as follows: Although the strong quenching effect of Co(II) and Ni(II) hampered the observation of an emission shift, their coordination to the probes **5** and **11** induced substantial absorption shifts (6-15 nm, Table S3). These results bolster the possibility of ratiometric detection of these metal ions based on AM-contact. We expect that this might be achieved by employing other tricyclic fluorophores which are resistant to fluorescence quenching by Co(II) and Ni(II).³⁴⁻³⁶

In our manuscript, we also demonstrated that AM-contact sensing can work with various fluorophores that emit fluorescence **across a broad wavelength region**. In particular, we achieved the ratiometric detection of the metal ions in the near-infrared region by using Si-pyronine. We believe that this is a significant achievement in the sensing research field as the development of ratiometric near-infrared probe for metal ions is still challenging task. Most of the ratiometric near-infrared probes for metal ions developed to date employ cyanine and BODIPY as the fluorophores, which operate with ICT as the sensing mechanism. To our knowledge, our result is the first example of Si-pyronine used for the ratiometric detection of metal ions. We would like to emphasize that this achievement was accomplished thanks to the high compatibility of AM-contact sensing with various fluorophores. This point was discussed in the discussion section of the revised manuscript (page 15) as follows: Of note, we successfully demonstrated that AM-contact sensing is operable with Si-pyronine, which emits a near-infrared fluorescence. This is a significant achievement in chemosensor research as the development of ratiometric near-infrared probes for metal ions is still a challenging task.³¹⁻³³ To our knowledge, this is the first example of Si-pyronine used in the ratiometric detection of metal ions.

In our differential metal ion sensing experiments, we successfully distinguished 8 metal ions using PCA analysis. We believe that the significance of this result can be found not only in the number of metal ions distinguished, but also in the novelty of the multicolor sensing system using AM-contact. As discussed in the main text (page 11), most of the differential metal ion sensing systems reported to date employ multi-well sensor assay systems, which require multiple steps and repeated fluorescence measurements. In contrast, our sensing system involves a single titration solution containing the set of AM-contact probes, which enables quick detection of the metal ions by a one-pot single fluorescence measurement. It is apparent that this achievement takes full advantage of the unique properties of AM-contact sensing, including high compatibility with various fluorophores and broad applicability to different metal ions. To more clearly state this point, we added the following sentence in the discussion (page 16): Of note, we have conveniently distinguished 8 metal ions by a one-pot single titration for differential sensing, which is difficult to achieve using the existing sensor array systems for differentiation of metal ions.²⁷⁻³⁰

Overall, this manuscript includes a number of the significant findings in many aspects, which broadly appeal to researchers in relevant fields. We thus believe that this manuscript meets the high novelty criteria required for publication in Communications Chemistry.

Comment 2: The ESI is very well organized but the actual NMR spectra of the new probes were not provided in the submission I reviewed.

Reply: In accordance with the comments, we added the NMR spectra of the new probes in the Supplementary Information.

Reviewers' comments:

Reviewer #1 (Remarks to the Author):

The authors suitably revised the manuscript, which is now ready for publication in Communications Chemistry.

Reviewer #2 (Remarks to the Author):

The authors have thoroughly addressed the reviewer comments, and I believe that this work is suitable for publication.

Reviewer #3 (Remarks to the Author):

The Authors have done an excellent job addressing the majority of the reviewers comments - including performing additional experiments where requested. The single concern now is that the NMR spectra - included at the request of reviewer 3 - do not convince the reader of the homogeneity of the structures proposed. Furthermore there are no ¹³C spectra?

For example the ¹H spectra of probes 5 and 9 are primarily solvent with exceptionally weak signals for the molecule of interest. The spectra are not properly labelled and appear to be scanned versions of the originals.

It appears there are two spectra for compound 11 yet the reader has no idea as to what they are. The bottom one does not register above baseline?

Until this aspect has been satisfactorily addressed the manuscript remains unsuitable for publication.

Reply to the comment of the reviewer 3

Comment: The Authors have done an excellent job addressing the majority of the reviewers comments - including performing additional experiments where requested. the single concern now is that the NMR spectra – included at the request of reviewer 3 – do not convince the reader of the homogeneity of the structures proposed. Furthermore there are no ^{13}C spectra?

For example the ^1H spectra of probes 5 and 9 are primarily solvent with exceptionally weak signals for the molecule of interest. The spectra are not properly labelled and appear to be scanned versions of the originals.

It appears there are two spectra for compound 11 yet the reader has no idea as to what they are. The bottom one does not register above baseline?

Until this aspect has been satisfactorily addressed the manuscript remains unsuitable for publication.

Reply: In accordance with the comment, we measured ^1H -NMR and ^{13}C -NMR of the six fluorescent probes. These spectral charts and chemical shift assignments are shown in Supplementary Material and highlighted by yellow marker. We believe that these new data clearly validate homogeneity of the probe structures we proposed.

REVIEWERS' COMMENTS:

Reviewer #3 (Remarks to the Author):

This is a very interesting manuscript. The Authors have now included ^1H and ^{13}C spectra and revised the characterization details in the ESI.

For compound 8 the spectra appear to be contaminated (an additional peak exists near 0 in the ^1H). Perhaps aggregation has occurred for 12 - maybe the choice of solvent CDCl_3 was not ideal and that has led to aggregation?

As the ^1H and ^{13}C spectra was the the key criticism from the previous round of reviewing a suitable explanation should be provided by the authors prior to publication. Are there related compounds where acquiring NMR has proven problematic? While not essential it would be ideal if new spectra in d_6 -DMSO could be acquired for 8 and 12 to minimize aggregation.

Reply to the comment of the reviewer 3

In accordance with the comment, we purified again probe **8** and measured the ^1H - and ^{13}C -NMR spectra in DMSO-d_6 . The new data were shown in the revised ESI (page S42 and S43). We found that the singlet noise peak around 0 ppm, which was observed in the previous ^1H -NMR spectrum, disappeared in the purified probe. We further confirmed that this noise peak also disappeared in the ^1H -NMR measurement using CDCl_3 , suggesting that the noise peak was derived from a small amount of contaminated material (silicone grease, etc.) and was not due to the aggregated probe.

In the case of probe **12**, we previously tried the ^1H -NMR measurement using DMSO-d_6 . However, we observed the broaden and rather complicated signals in the ^1H -NMR spectrum. We assumed that this unwanted result were due to the slow conformational equilibrium of probe **12** (probably at its flexible ligand site) in DMSO-d_6 . We therefore tested other solvents and found that CDCl_3 is the best one that affords the sharper ^1H -NMR signals, which support the proposed structure of probe **12**. We think that the rather broaden signals in the ^1H -NMR (page S50) would be due to the conformational equilibrium of probe **12**. Meanwhile, the sharp signals in the ^{13}C -NMR (page S51) suggest that probe **12** was well soluble in CDCl_3 and did not form aggregation.